# A neurogenic signature involving monoamine Oxidase-A controls human thermogenic adipose tissue development

Javier Solivan-Rivera[1], Zinger Yang Loureiro[1], Tiffany DeSouza[2], Anand Desai[2], Sabine Pallat[1], Qin Yang[1], Raziel Rojas-Rodriguez[2], Rachel Ziegler[2], Pantos Skritakis[2], Shannon Joyce[2], Denise Zhong[2], Tammy Nguyen[3,4], Silvia Corvera[2,4]*

[1]Morningside Graduate School of Biomedical Sciences, University of Massachusetts Medical School, Worcester, United States; [2]Program in Molecular Medicine, University of Massachusetts Medical School, Worcester, United States; [3]Department of Surgery, University of Massachusetts Medical School, Worcester, United States; [4]Diabetes Center of Excellence, University of Massachusetts Medical Center, Worcester, United States

*For correspondence: silvia.corvera@umassmed.edu

Competing interest: The authors declare that no competing interests exist.

**Abstract** Mechanisms that control 'beige/brite' thermogenic adipose tissue development may be harnessed to improve human metabolic health. To define these mechanisms, we developed a species-hybrid model in which human mesenchymal progenitor cells were used to develop white or thermogenic/beige adipose tissue in mice. The hybrid adipose tissue developed distinctive features of human adipose tissue, such as larger adipocyte size, despite its neurovascular architecture being entirely of murine origin. Thermogenic adipose tissue recruited a denser, qualitatively distinct vascular network, differing in genes mapping to circadian rhythm pathways, and denser sympathetic innervation. The enhanced thermogenic neurovascular network was associated with human adipocyte expression of THBS4, TNC, NTRK3, and SPARCL1, which enhance neurogenesis, and decreased expression of MAOA and ACHE, which control neurotransmitter tone. Systemic inhibition of MAOA, which is present in human but absent in mouse adipocytes, induced browning of human but not mouse adipose tissue, revealing the physiological relevance of this pathway. Our results reveal species-specific cell type dependencies controlling the development of thermogenic adipose tissue and point to human adipocyte MAOA as a potential target for metabolic disease therapy.

## Editor's evaluation

This paper provides important new information that will be of wide interest in the fields of metabolism and diabetes research. The authors implanted human adipocyte progenitors is potentially a novel approach to analyze the development of human thermogenic adipose tissue.

## Introduction

A positive association between thermogenic 'beige/brite' adipose tissue and metabolic health in humans has been repeatedly observed (*Cypess et al., 2009*; *Enerbäck, 2010*; *van Marken Lichtenbelt et al., 2009*; *Virtanen et al., 2009*), and multiple studies have shown that human thermogenic adipocytes transplanted into mice affect whole-body metabolism (*Min et al., 2016*; *Tsagkaraki et al., 2021*; *Wang et al., 2020*). These studies provide evidence for a cause-effect relationship and a rationale for enhancing the activity and/or abundance of thermogenic adipose tissue as a therapeutic

approach to metabolic disease. However, the cellular and molecular mechanisms by which human thermogenic adipose tissue develops and is maintained are largely unknown.

In adults, thermogenic adipose tissue develops in response to chronic cold exposure, extensive skin burns, and catecholamine producing tumors (*Cypess et al., 2012*; *Cypess et al., 2015*; *Dong et al., 2014*; *Sidossis et al., 2015*; *van der Lans et al., 2013*). Formation of new thermogenic adipose tissue involves the generation of new adipocytes expressing the characteristic mitochondrial uncoupling protein 1 (UCP1) of new vasculature to facilitate oxygen consumption and heat dissipation (*Cao et al., 2019*; *Chi et al., 2018*; *François et al., 2019*; *Sievers et al., 2020*), and of new innervation to mediate sympathetic signaling (*Cao et al., 2019*). Signaling between multiple cell types is likely to be involved in the process of tissue expansion; for example, adipocytes secrete factors that induce vascularization and innervation (*Zhao et al., 2018*), catecholamines can stimulate angiogenesis, and progenitor cells niched in the newly forming vasculature can differentiate into thermogenic adipocytes (*Lee et al., 2015*; *Min et al., 2019*; *Shao et al., 2019*). The concurrence of adipogenesis, angiogenesis and innervation make it difficult to determine cell-type-specific mechanisms and temporal interdependencies, and to identify those mechanisms that could be harnessed to enhance thermogenic adipose tissue mass or activity. Understanding mechanisms of human thermogenic adipose tissue expansion is particularly difficult due to limited models available to study human tissue development. Our laboratory has developed methods to obtain large numbers of mesenchymal progenitor cells from human adipose tissue and shown that these can give rise to multiple adipocyte subtypes (*Min et al., 2019*). Moreover, these cells can be implanted into immunocompromised mice, where they develop metabolically integrated adipose tissue (*Rojas-Rodriguez et al., 2019*). We reasoned that this cell implantation model, by clearly separating the adipocyte component (human), from vascular and neuronal components (mouse), could help deconvolve mechanisms of development and maintenance of human thermogenic adipose tissue.

Here, we report that human adipose tissue generated in mice maintains species-specific features, including human adipocyte cell size, thermogenic gene expression, and the capacity to recruit a mature neurovascular network. Thermogenic adipocytes recruit a denser neurovascular network with distinct transcriptomic characteristics, suggesting features of the adipose tissue vasculature that may be important for thermogenic adipose tissue formation and function. Our studies also suggest mechanisms by which thermogenic adipocytes can recruit a distinct neuro-vasculature, as we find they differentially express genes involved in paracrine signaling of neurogenesis, and in modulation of neurotransmitter tone. Notably, *THBS4* and *SPARCL1*, which directly stimulate synapse formation (*Gan and Südhof, 2019*), are the most highly differentially expressed genes distinguishing thermogenic from non-thermogenic adipocytes in the hybrid tissue. In addition, we find that MAOA, which catalyzes the oxidative deamination and inactivation of multiple neurotransmitters (*Tipton, 2018*), is robustly expressed in human adipocytes and in adipocytes within hybrid tissue, but is suppressed during the formation of thermogenic adipose tissue. Functionally, systemic inhibition of MAOA with the irreversible inhibitor clorgyline results in enhanced expression of *UCP1* in adipocytes within the hybrid human adipose tissue, but not in endogenous mouse adipose tissue. These data reveal important species-distinct mechanisms controlling the establishment and maintenance of human adipocyte thermogenic phenotype in vivo.

## Results

### Formation of functional hybrid adipose depots from human cells

To generate cells for implantation we derived mesenchymal progenitor cells from human subcutaneous adipose tissue (*Rojas-Rodriguez et al., 2019*) as detailed in Materials and methods, and injected cell suspensions subcutaneously into each flank of nude mice (*Figure 1—figure supplement 1A, B*). Discernible adipose tissue pads, and circulating human adiponectin, were detected in all mice (*Figure 1—figure supplement 1C, D*), demonstrating that human adipocytes were able to develop into hybrid tissue and secrete characteristic adipokines for an extended period after implantation. To monitor adipocyte development, we performed immunohistochemistry of the excised tissue with an antibody to Perilipin 1 (PLIN1), which delineates lipid droplets. This analysis revealed a progressive increase in cell size over a period of 16 weeks (*Figure 1A*). Notably, human adipocytes developed from implanted cells exceeded the size of mouse endogenous subcutaneous white adipocytes as

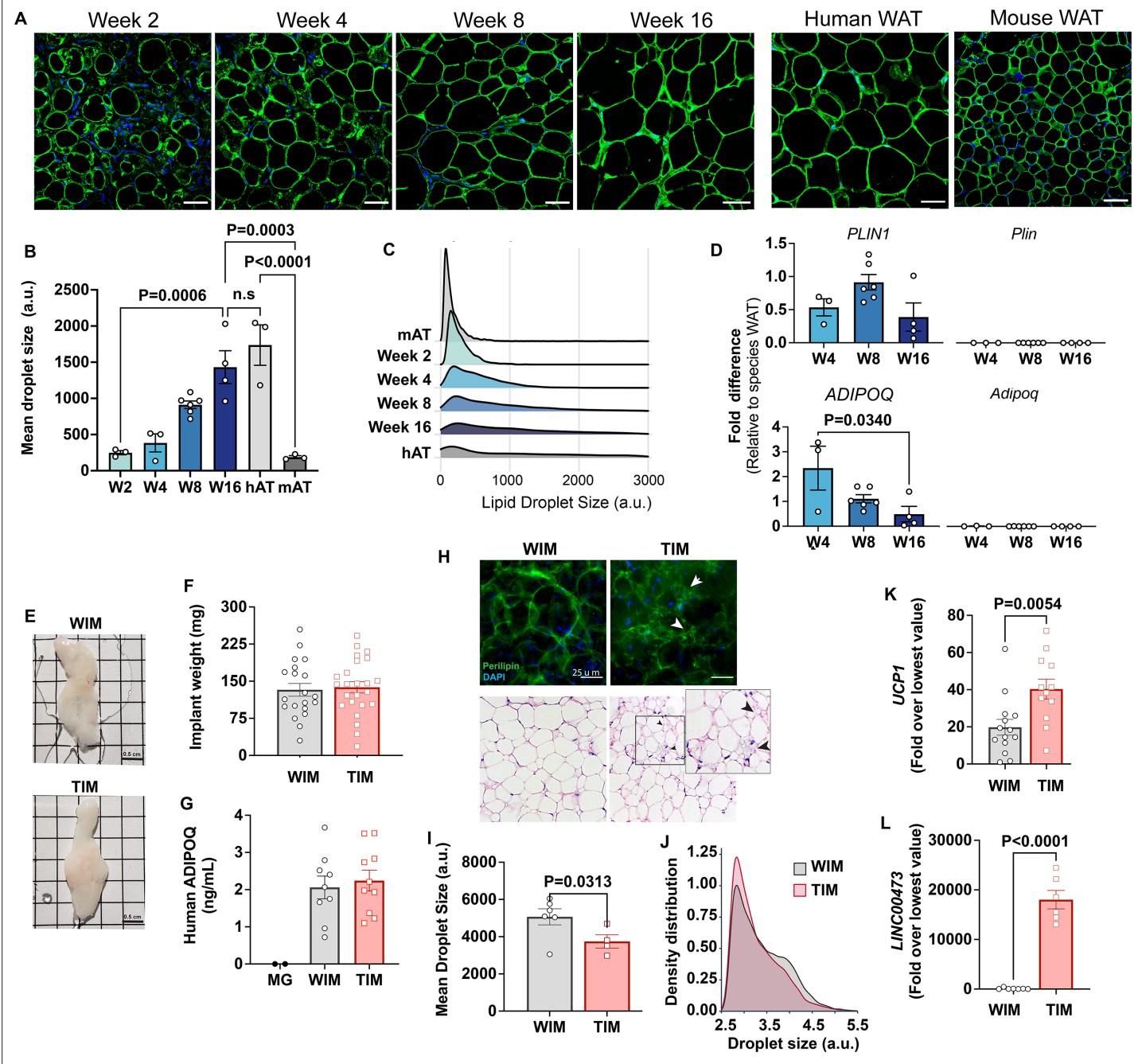

**Figure 1.** Characteristics of adipocytes during development of WIM and TIM. (**A**) Representative images of PLIN1 (green) and DAPI (blue) staining of WIM at the indicated weeks after implantation, and of inguinal mouse and subcutaneous human adipose tissue, scale bar = 50 μm. (**B**) Quantification of droplet size of PLIN1-stained implants and tissues. Each symbol represents the mean area of all cells within three random fields of a single implant or tissue section. Bars represent the means and SEM of n=3 n=3, n=6, n=4, implants at weeks 2, 4, 8, and 16 respectively, and n=3 for independent human or mouse adipose tissue samples. Statistical significance of the differences between samples was calculated using one-way ANOVA with Tukey's adjustment for multiple comparisons, and exact p-values are shown. (**C**) Density distribution plot of droplet size of developing implants and control tissue. Each density distribution represents the combined density of all samples for each group (n=3–6). (**D**) RT-PCR of excised implants using species- specific probes for genes indicated. Values were normalized to those obtained from human or mouse adipose tissue samples exemplified in A. Symbols are the means of technical duplicates for each implant, and bars are the means and SEM of n=3, n=6 and n=4 implants at weeks 4, 8, and 16, respectively. Statistical significance of the differences between samples was calculated using one-way ANOVA with Tukey's adjustment for multiple comparisons, and exact p-values are shown. (**E**) Representative images of white implants (WIM) and thermogenic implants (TIM) at 8 weeks of development, grid size = 0.5 cm. (**F**) Weight in mg of each excised implant from bi-laterally implanted mice. Bars are the mean and SEM of n=20 and n=24 implants for WIM and TIM, respectively. (**G**) Circulating human-specific adiponectin (ng/ml) from bilaterally implanted mice. Symbols are the

*Figure 1 continued on next page*

*Figure 1 continued*

mean of technical duplicates for n=2, n=9, and n=10 mice implanted with Matrigel alone (MG), WIM or TIM, respectively. (**H**) Whole mounts of WIM and TIM at 8 weeks after implantation stained for PLIN1 (top panels), or with H&E. Arrowheads highlight clusters of dense, multilocular droplets in TIM. (**I**) Mean droplet size (arbitrary units) in WIM and TIM. Symbol represent the mean areas of all cells from two independent sections from each implant. Bars represent the mean and SEM of n=6 and n=4 WIM and TIM implants, respectively. Statistical significance of the difference was calculated using unpaired, one-tailed Student t-test and resulting p-value is shown. (**J**) Density distribution plot of droplet size of WIM and TIM. Each density distribution represents the average density of all samples for each group. (**K, L**) RT-PCR of excised implants using human-specific probes for the genes indicated. Symbols are the means of technical duplicates for each implant, and bars are the means and SEM of n=14 and n=12 for *UCP1*, and n=7 and n=6 for *LINC00473*, in WIM and TIM, respectively. Values were normalized to the lowest expression level in the entire data set of n=26 implants. Statistical significance of the differences was calculated using unpaired, two-tailed Student t-tests, and exact p-values are shown.

The online version of this article includes the following figure supplement(s) for figure 1:

**Figure supplement 1.** Development of implants from cultured adipocytes in mice.

early as week 4 after implantation and continued to enlarge to match the mean size of adipocytes in the human tissue from which progenitor cells were obtained (*Figure 1A and B*). The change in mean adipocyte size was mostly attributable to a shift from multilocular to unilocular adipocytes (*Figure 1C*), suggesting that as lipid droplets enlarge during development they also coalesce. Expression of human *PLIN1* was relatively stable from weeks 4 to 16 (*Figure 1D*, top panels), but *ADIPOQ* expression decreased as adipocyte size increased (*Figure 1D*, lower panels), mimicking the known inverse correlation between ADIPOQ production and adipocyte hypertrophy (*Robciuc et al., 2016*). Importantly, there was no detectable expression of mouse *Plin1* or *Adipoq*, in the hybrid tissue (*Figure 1D*, right top and lower panels), indicating that adipocytes in the developed depots are exclusively of human origin. These results indicate that adipocytes continue to develop in the mouse while retaining fundamental features of human adipose tissue, such as large adipocyte size and adiponectin expression.

We then asked whether depots formed from white or thermogenically induced adipocytes would differ from each other. Mice were implanted with cells that were treated with vehicle (White IMplanted = WIM) or forskolin (Thermogenic IMplanted = TIM), and developed depots analyzed after 8 weeks of implantation. The macroscopic features and weight of TIM and WIM were similar (*Figure 1E and F*), as was the average concentration of circulating human adiponectin in mice harboring TIM compared to WIM (*Figure 1G*). However, multilocular adipocytes could be detected in wholemount staining of TIM, by either PLIN1 immunostaining of whole mounts (*Figure 1H*, top panels, white arrows) or by H&E staining of tissue sections (*Figure 1H*, lower panels, black arrows). Image quantification revealed a significant decrease in mean droplet size in TIM (*Figure 1I*), resulting from a higher density of small droplets (*Figure 1J*), which are typically seen in thermogenic adipocytes. Expression of human UCP1 was higher in TIM (*Figure 1K*), and expression of LINC00473, a log non-coding RNA which is detected specifically in human thermogenic adipocytes (*Tran et al., 2020*), was exclusively detected in TIM (*Figure 1L*). These results indicate that thermogenic features of implanted cells are maintained for a minimum of 8 weeks in vivo during which adipose tissue continues to develop from implanted cells.

## Beige adipocytes develop into adipose tissue with a denser neurovascular architecture

Thermogenic adipose tissue in mice is characterized by higher vascular and innervation density (*Chi et al., 2018*; *Chi et al., 2021*). To determine whether WIM and TIM development was accompanied by differences in vascularization we stained whole-mounts and thin sections from excised implants with endothelial cell stain isolectin (*Figure 2A*). We first analyzed the dynamics of vascular development in WIM over time after implantation. We detected dispersed endothelial cells as early as week 2 in whole-mount specimens (*Figure 2A*, top panels), and between weeks 4 and 8, a well-developed vascular network, closely resembling that seen in human adipose tissue, was observed. Vessels formed in apposition to enlarging adipocytes, as seen in sections co-stained with isolectin and antibodies to PLIN1 (*Figure 2A*, lower panels). The total intensity of isolectin staining, relative to PLIN1 staining to account for adipocyte mass, did not differ over time (*Figure 2B*). However, the mean size of vascular structures significantly increased between weeks 4 and 8 (*Figure 2C*), corresponding with the appearance of discernible vessels. Expression of mouse vascular endothelial cadherin (*Cdh5*) did not differ over time, consistent with the intensity of isolectin staining (*Figure 2D*, left panel), and was exclusively of mouse origin as human *CDH5* was undetectable (*Figure 2D*, right panel). These results indicate

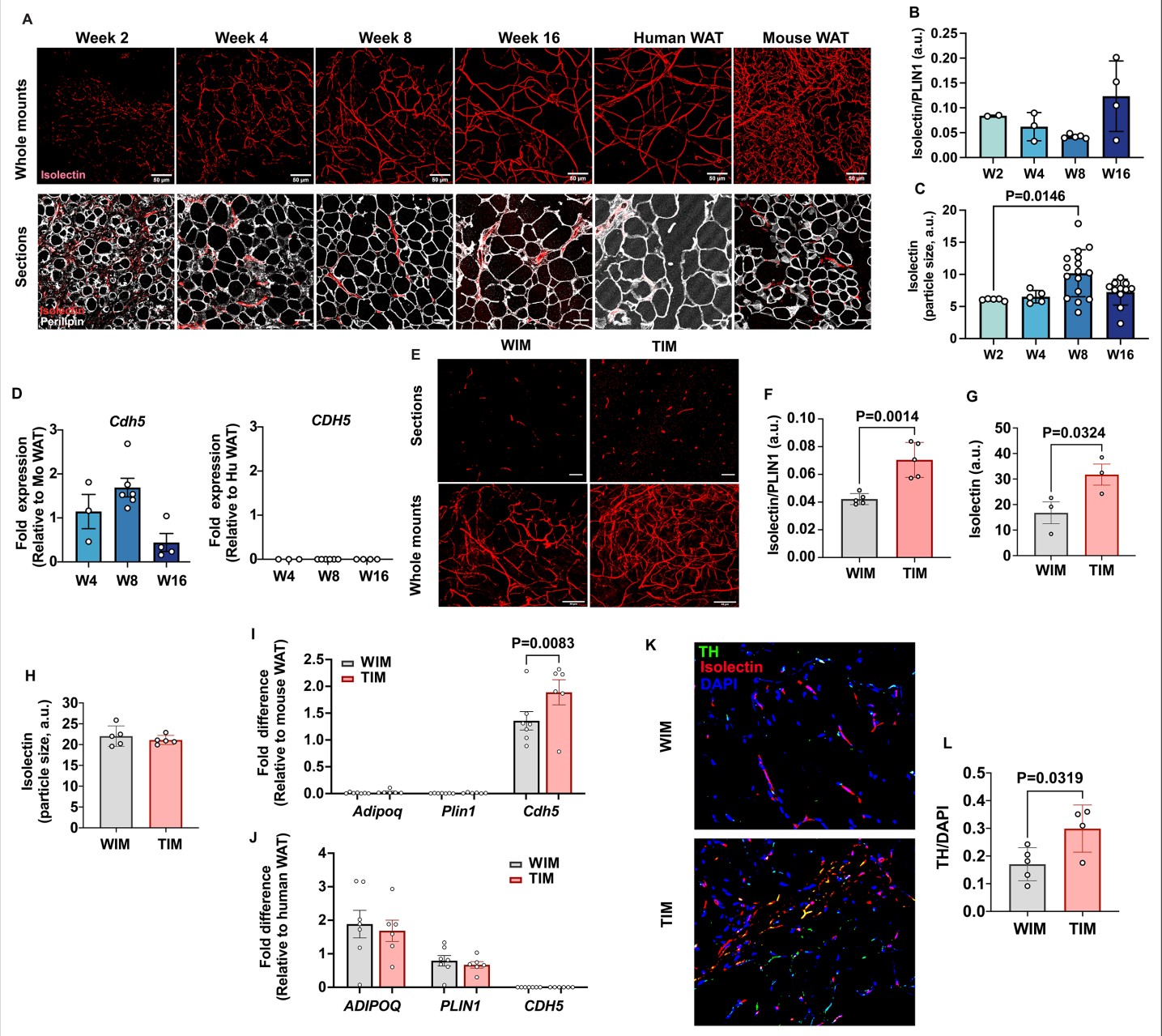

**Figure 2.** Vascular development of WIM and TIM. (**A**) Representative images of isolectin staining of WIM at the indicated weeks after implantation, and of inguinal mouse and subcutaneous human adipose tissue, scale bar = 50 µm. (**B**) Total isolectin staining in implants. Each symbol represents the sum of the areas of particles stained by isolectin, normalized to the total area stained by Plin1, of three random fields from a single implant. Bars represent the means and SEM of n=2, n=3, n=5, n=4, implants at weeks 2, 4, 8, and 16, respectively. C. Mean size of isolectin-stained particles. Each symbol represents the mean area of particles stained by isolectin in three random fields per implant. Bars represent the means and SEM of n=5, n=5, n=16, n=11, implants at weeks 2, 4, 8, and 16, respectively. Significance of the differences between samples was calculated using one-way ANOVA with Tukey's adjustment for multiple comparisons, and exact P-values are shown. (**D**) RT-PCR of excised implants using species-specific probes for *CDH5/Cdh5*. Values were normalized to those obtained from mouse adipose tissue samples. Symbols are the means of technical duplicates for each implant, and bars are the means and SEM of n=3, n=6, and n=4 implants at weeks 4, 8, and 16 respectively. (**E**) Representative images of thin sections (upper panels) and whole mounts (lower panels) of WIM and TIM after 8 weeks of implantation, stained with isolectin. (**F, G**) Total isolectin staining of thin sections and whole mounts, assessed as in (**B**). Bars represent the means and SEM of n=5, implants for thin sections, and n=3 for whole mounts of WIM and IM, respectively. Significance of the differences between samples was calculated using unpaired two-tailed Student t-tests. (**H**) Mean size of isolectin-stained particles in WIM and TIM, analyzed as in (**C**). Bars represent the means and SEM of n=5, implants. (**I, J**) RT-PCR of excised implants using mouse-specific (**I**) and human-specific (**J**) probes for the genes indicated. Values were normalized to those obtained from mouse and human adipose tissue samples. Symbols are the means of technical duplicates for each implant, and bars are the means and SEM of n=7, and n=6 WIN and TIM implants respectively.

*Figure 2 continued on next page*

*Figure 2 continued*

Significance of the differences between WIM and TIM for each gene was calculated using one-way ANOVA with the Holm-Sidak adjustment for multiple comparisons, and exact p-values are shown. (**K**) Representative images of thin sections WIM and TIM after 8 weeks of implantation, stained for isolectin (red), tyrosine hydroxylase (TH, green) and DAPI (blue). (**L**) Total TH staining, where bars represent the means and SEM of n=5, and n=4 sections of WIM and TIM, respectively. Significance of the differences between samples was calculated using un-paired two-tailed Student t-test and the exact p-value is shown.

The online version of this article includes the following figure supplement(s) for figure 2:

**Figure supplement 1.** Density of macrophages in WIM and TIM.

that endothelial cells invade the emerging tissue early in development, and subsequently mature into morphologically discernable vessels. Even though all vasculature was of mouse origin, the vascular network was morphologically indistinguishable from human WAT (*Figure 2A*, comparing Week 16 with human adipose tissue [Human AT]).

We then compared the vasculature between TIM and WIM after 8 weeks of implantation in thin sections (*Figure 2E*, upper panels) and whole-mount fragments (*Figure 2E*, lower panels). Total intensity of isolectin staining relative to PLIN1 staining was increased in TIM (*Figure 2F*), as was the non-normalized intensity of isolectin in whole-mount specimens (*Figure 2G*). The overall size of vascular structures was not different between WIM and TIM (*Figure 2H*), suggesting that vessel maturation was unaffected. Gene expression levels of *Cdh5* were higher in TIM (*Figure 2I*), but those of human adipocyte genes ADIPOQ or PLIN1, were not different between TIM and WIM (*Figure 2J*). These results indicate that thermogenic adipocytes induce the formation of a denser vascular network compared to non-thermogenic cells. We next asked whether TIM would have higher content of blood cells, which we would expect if the newly formed vascular network was functional. For this we measured the hematopoietic cell marker CD45 and the macrophage marker F4/80 (*Figure 2—figure supplement 1*). We find that TIM had significantly higher number of macrophages compared to WIM, proportional to the increase in vasculature, and consistent with a denser, functional vascular network.

Sympathetic innervation often accompanies the development of vasculature (*Carmeliet, 2003*; *Pellegrinelli et al., 2018*; *Reinert et al., 2014*). To determine whether development of TIM was accompanied by sympathetic nerve recruitment, we imaged implants for tyrosine hydroxylase (TH) and isolectin. Punctate staining consistent with nerve terminals was detected in all implants, with some overlapping with isolectin (*Figure 2K*). Quantification revealed significantly higher TH staining in TIM compared to WIM (*Figure 2L*), indicating enhanced sympathetic recruitment in conjunction with enhanced vascularization.

## Development and maintenance of TIM thermogenic phenotype is not cell autonomous

To determine whether maintenance of the thermogenic phenotype in-vivo depends on the neurovascular network or is an autonomous property of the cells, we examined the capacity of adipocytes to retain thermogenic gene expression in culture. We compared adipocytes that were never stimulated to adipocytes that were chronically stimulated with Forskolin (Fsk), and with adipocytes from which Fsk was withdrawn after a period of chronic stimulation (*Figure 3A*). Removal of Fsk from chronically stimulated adipocytes resulted in a rapid increase in droplet size, which reached the size seen in cells that were never exposed to Fsk within days of removal (*Figure 3B and C*). Forskolin withdrawal also resulted in a sharp decrease in *UCP1* (*Figure 3D*) and *LINC00473* (*Figure 3E*) expression, with values becoming statistically non-different from non-induced adipocytes within 5 days of withdrawal. Interestingly, the expression of *LINC00473* decreased even in the presence of chronic Fsk stimulation, while *UCP1* levels continued to increase over time, revealing different pathways of adaptation to chronic stimulation. Irrespective of these gene-specific differences, these results indicate that the thermogenic phenotype of adipocytes in vitro is completely reversed within days of withdrawal of stimulation. We conclude that adipocytes lack the capacity to autonomously maintain a thermogenic phenotype, and that vascularization and innervation generated following implantation are likely to play a critical role in sustaining their thermogenic properties.

We then examined whether implanted adipocytes might have systemic effects that would impact host adipose depot thermogenic state. The weights of mouse inguinal white and interscapular brown adipose tissue (iWAT and iBAT, respectively) were similar between mice implanted with either WIM

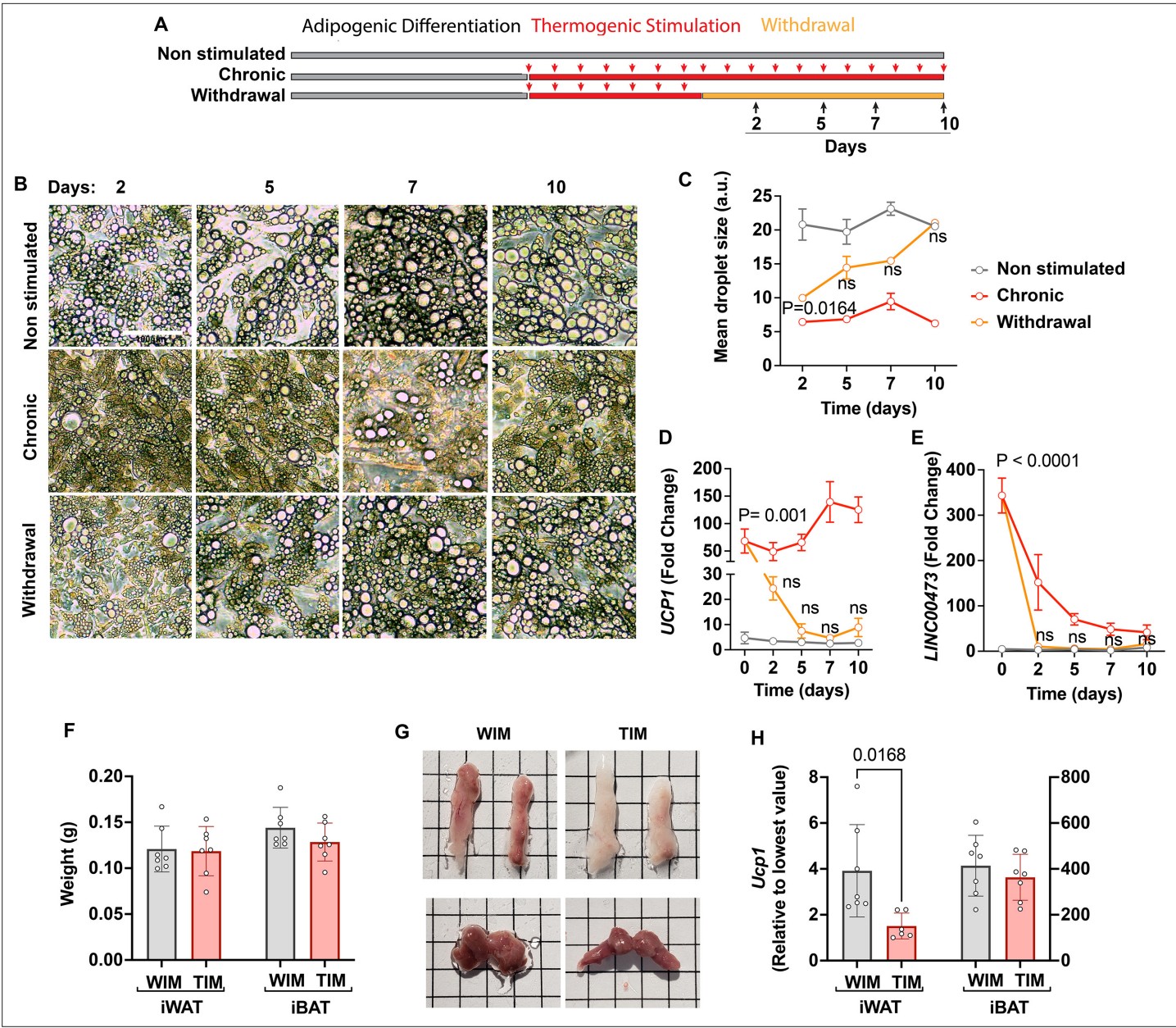

**Figure 3.** Capacity of thermogenic adipocytes to autonomously maintain their phenotype, and their effect on host adipose depots. (**A**) Experimental paradigm for in vitro experiments. (**B**) Phase images of cultured adipocytes during chronic stimulation and withdrawal. (**C**) Quantification of droplet size, where symbols are the mean and SEM of all droplets in single images from each of n=3 independent culture wells. (**D, E**) RT-PCR with human-specific probes for UCP1 (**D**) and LINC00473 (**E**), where symbols are the mean and error bars the SEM of n=3 independent culture wells assayed in technical duplicate. Values are expressed as the fold difference relative to the lowest value in the dataset. For (**C**), (**D**) and (**E**), statistical significance of differences at each time point between the withdrawal and the non-induced group, was measured using two-way ANOVA with Dunnet's multiple comparison test, and exact p-values are shown. (**F**) Weight of excised inguinal (iWAT) and interscapular (iBAT) after 8 weeks of implantation with WIM or TIM. Symbols correspond to the summed weight of two inguinal or the single interscapular pad per mouse and bars the mean and SEM of n=7 mice. (**G**) Representative images of excised fat pads. Top panels, iWAT; bottom panels iBAT. H. RT-PCR of *Ucp1*, where each symbol is the mean of technical duplicates, and bars represent the mean and SEM of n=6–7 pads. Data are expressed as the fold difference relative to the lowest value in the set, with scales for iWAT and iBAT on the left and right Y axes, respectively. Statistical significance of differences was measured using non-paired, two-tailed Student t-tests and exact P-values are shown.

or TIM (*Figure 3F*), but the iWAT, which in athymic nude mice is noticeably beige, looked 'whiter' in mice harboring TIM compared to WIM (*Figure 3G*). This morphological difference was associated with lower *Ucp1* expression in iWAT of TIM-implanted mice, while no differences were seen in iBAT (*Figure 3H*). These results indicate that hybrid thermogenic depots influence endogenous thermogenic adipose tissues, potentially through similar feedback mechanisms that take place when mouse thermogenic tissues are transplanted into mice (*Schulz et al., 2013*).

## Distinct gene expression features of adipocytes and host cells in WIM and TIM

To explore underlying mechanisms by which WIM and TIM elicit differential vascularization and innervation, we performed bulk RNA sequencing after 8 weeks of implantation. To distinguish transcriptomes corresponding to either mouse or human cells, fastq files were aligned to both genomes, and resulting reads were classified as either of human or mouse origin using the XenofilteR R-package, and we then used mouse genes to infer the mouse cell types that comprise WIM and TIM (*Figure 4A*). As a framework we used single-cell datasets from *Burl et al., 2018*, who profiled stromovascular cells from both inguinal and gonadal mouse depots. Consistent with their findings, we distinguish 9 clusters, corresponding to adipose stem cells at distinct stages of differentiation, vascular endothelial cells, vascular smooth muscle cells, lymphocytes, dendritic cells, macrophages, neutrophils, and an unidentified cluster (*Figure 4B*). The most prominent host-derived cell populations in WIM and TIM, predicted by dampened weighted least squares deconvolution (DWLS, *Tsoucas et al., 2019*; https://github.com/dtsoucas/DWLS; *Tsoucas, 2021*), were adipose stem/progenitor cells, macrophages, and vascular endothelial cells, with vascular smooth muscle cells, dendritic cells, and other immune cells at very low numbers. No detectable differences in the relative proportion of mouse cell types within WIM and TIM were observed (*Figure 4C*). It is interesting to note that mouse progenitor cells infiltrating the developing tissue did not differentiate into adipocytes, as the expression of mouse *Plin1* and *Adipoq* in WIM and TIM is negligible (*Figure 2I*).

We then probed for differences in expression levels of mouse genes between WIM and TIM. Of 13,036 detected genes, 86 were higher in TIM (range 1.25- to 414-fold), and 124 genes were higher in WIM (range 1.25- to 8-fold), with a p-adjusted value lower than 0.05 (*Figure 4D* and *Supplementary file 1*). Pathway analysis of differentially expressed genes (*Figure 4D*) revealed circadian clock genes as the most prominent pathway, with higher expression in TIM of *Npas2* and *Arntl* (3.9- and 3.5-fold higher in TIM, padj values 9.39E-05 and 7.36E-06, respectively) and higher expression in WIM of *Dbp*, *Ciart*, *Per2,* and *Per3* (3.1-, 2.9-, 2.8-, and 2.4-fold, padj values 1.28E-20, 0.008, 1.22E-11, and 4.99E-08 respectively). We also find enrichment in cholesterol biosynthetic and extracellular matrix remodeling pathways (*Figure 4E*). We then asked if expressed genes correspond to a specific cell type, by searching for their expression in stromovascular cells from both inguinal and gonadal mouse depots shown in *Figure 4B*. Differentially expressed genes associated with circadian rhythms (e.g. *Dbp*, *Per3*) and extracellular matrix remodeling (e.g. *Adamts4*, *Col4a2*) could be detected within clusters corresponding to progenitor cells and vascular endothelial cells (*Figure 4F*). Genes in the cholesterol biosynthesis pathway (*Fdps* and *Cyp51*) were seen in clusters corresponding to dendritic cells and macrophages. These results suggest that mouse progenitor and endothelial cells respond to signals emanating from human thermogenic versus non-thermogenic adipocytes to adopt distinct functional phenotypes.

To elucidate how human thermogenic adipocytes induce differential vascularization and innervation, we searched for genes differentially expressed between WIM and TIM that mapped to the human genome. Of 9655 detected genes, 515 were higher in WIM and 218 were higher in TIM, with a p-adjusted value lower than 0.05 (*Figure 4G*, *Supplementary file 2*). The most highly differentially expressed genes were associated with neurogenesis, including higher expression in TIM of *THBS4*, *TNC*, *SPARCL1,* and *NTRK3* (5.6-, 2.7-, 2.7-, and 2.4- fold, padj-values 0.006, 0.0003, 9.3E-09, and 0.0001, respectively) (*Reinert et al., 2014*; *Schulz et al., 2013*; *Burl et al., 2018*; *Boyle et al., 2004*; *Camell et al., 2017*; *Pirzgalska et al., 2017*). Pathway analysis of all genes with higher expression in TIM (*Figure 4H*) revealed enrichment in ribosomal protein-encoding genes, suggesting higher protein synthesis activity, followed by pathways affecting extracellular matrix composition, including numerous collagens. Many of these genes have been implicated in the control of blood vessel development, and could underlie the increased vascular density seen in TIM. As expected, *UCP1* mean values were

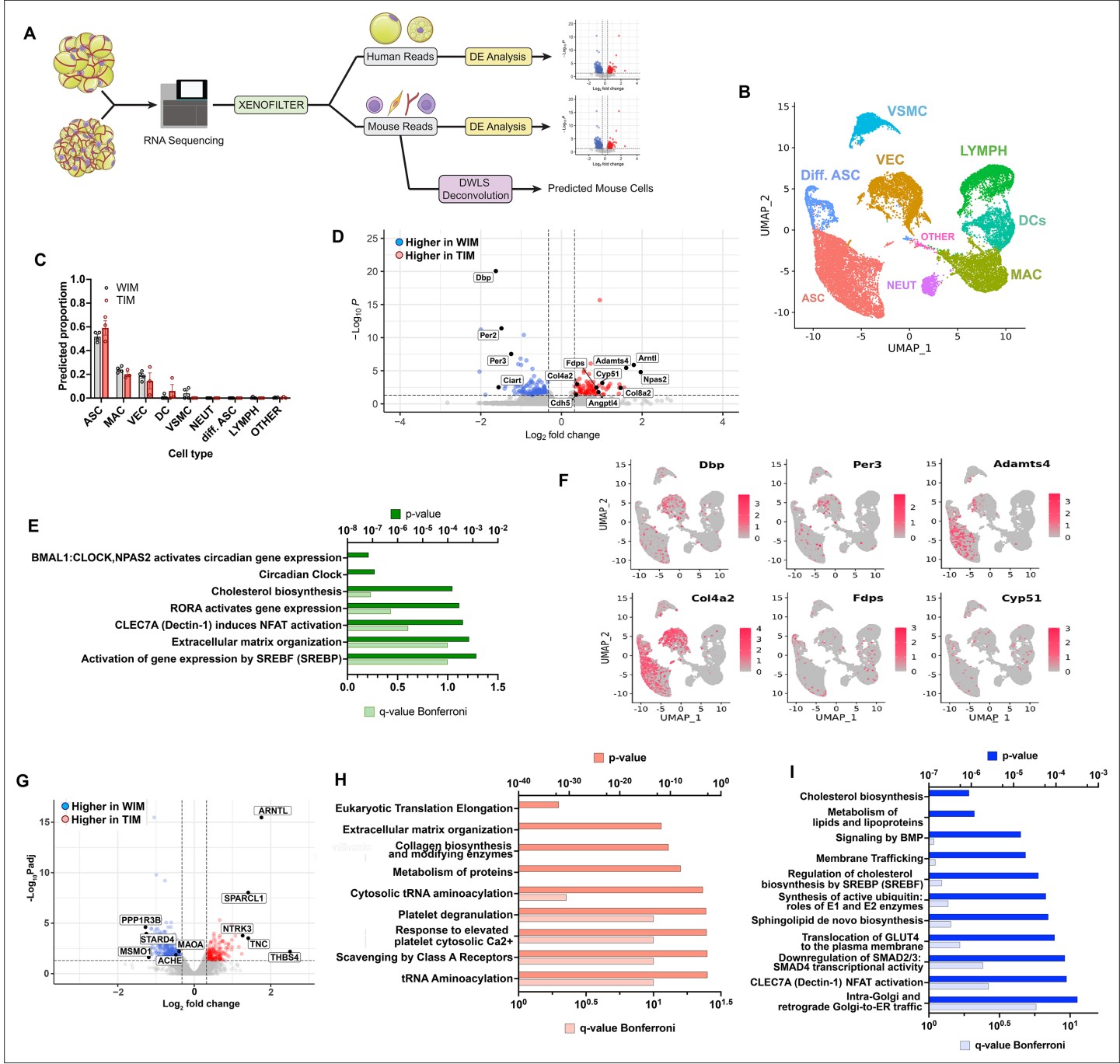

**Figure 4.** Distinct gene expression signatures of adipocytes and stromal cells in WIM and TIM. (**A**) Overview of analysis paradigm. (**B**) Dimensional reduction plot of data from **Burl et al., 2018**, using Uniform Manifold Approximation and Projection (UMAP), where ASC = adipose stem cells, Diff. ASC = differentiating adipose stem cells, VEC = vascular endothelial cells, VSMC = Vascular smooth muscle cells, LYMPH = Lymphocytes, DCs = Dendritic cells, MAC = macrophages, NEUT = Neutrophils, OTHER = unidentified cluster. (**C**) Predicted proportion of cells present in WIM and TIM. Symbols correspond to data from an individual implant and bars the mean and SEM of n=4 and n=3 WIM and TIM respectively. (**D**) Volcano plot of genes mapping to the mouse genome which are differentially expressed between WIM and TIM. (E) Pathway (REACTOME) enrichment analysis of mouse genes differentially expressed between WIM and TIM. (**F**) Mouse genes differentially expressed between WIM and TIM mapped onto the UMAP representation shown in (A), with cells expressing the indicated genes highlighted in red. (**G**) Volcano plot of genes mapping to the human genome differentially expressed between WIM and TIM. (**H,I**) Pathway (REACTOME) enrichment analyses of human genes more highly expressed in TIM (**H**) or WIM (**I**).

higher in TIM (18.2 and 42.3 in WIM and TIM, respectively), consistent with the significantly elevated *UCP1* expression seen by RT-PCR (*Figure 1K*). Genes expressed at higher levels in WIM were associated with cholesterol and lipid metabolism as well as insulin action (*Figure 4I*), consistent with a more prominent role for lipid storage pathways in non-thermogenic adipocytes as they develop in vivo.

To identify genes that might be most relevant to the maintenance of the thermogenic phenotype, we searched for genes differentially expressed between non-thermogenic and thermogenic adipocytes in culture prior to implantation that remain differentially expressed 8 weeks after implantation. 988 genes were more highly expressed in thermogenic adipocytes, and 729 more highly expressed in non-thermogenic adipocytes (>twofold, p- adjusted values <0.05; *Figure 5A*, *Supplementary file 3*). Genes were assigned to Gene Ontology (GO) categories using https://go.princeton.edu/cgi-bin/GOTermMapper, a tool for mapping granular GO annotations to a set of broader, high-level GO parent terms (*Boyle et al., 2004*). Genes more highly expressed in thermogenic adipocytes disproportionally mapped to GO categories associated with secretion (*Figure 5B*) and were enriched in pathways of extracellular matrix organization and signaling through GPCRs and IGFs (*Figure 5C*). Genes that were up regulated in non-thermogenic adipocytes were enriched in lipid metabolism pathways and transcriptional control of adipogenesis (*Figure 5D*). Of the 988 genes that were more highly expressed in thermogenic adipocytes prior to implantation, 21 remained more highly expressed in TIM compared to WIM after 8 weeks (*Supplementary file 4*), and 6 of these (*MMP14*, *ADAM19*, *TNC*, *ELN*, *COL12A1*, and *TGFB3*) are associated with extracellular matrix organization, which was the only pathway enriched by these genes (*Figure 5E*). Of the 729 genes that were expressed at higher levels in non-thermogenic adipocytes prior to implantation, 28 remained higher in WIM compared to TIM (*Supplementary file 4*). Pathway analysis of these genes revealed neurotransmitter clearance as the top enriched pathway, with two genes, *ACHE* and *MAOA*, being higher in WIM.

A lower expression of *ACHE* and *MAOA* in adipocytes would be expected to enhance neurotransmitter tone and maintenance of the thermogenic phenotype. Indeed, in mouse models, Maoa activity has been implicated in the regulation of norepinephrine stimulated adipose tissue lipolysis and browning, but its expression has been localized to a specific macrophage population (*Camell et al., 2017*; *Pirzgalska et al., 2017*; *Rogers et al., 2012*). Our finding of *MAOA* transcripts in WIM and TIM suggested expression in human adipocytes. To compare the levels of *MAOA* expression between species, we leveraged a recently published dataset that harmonizes single nuclei and single cell sequencing of mouse and human subcutaneous and visceral adipose tissues to generate a comprehensive atlas of adipose tissues transcripts (*Emont et al., 2022*). While single-cell and single-nuclei transcriptomics are biased to detect more highly expressed genes, the approach allows reasonable comparison between cells within a species. In human adipose tissues, *MAOA* transcripts (*Figure 6A*, top row) were clearly detected and abundant in the population of cells expressing *ADIPOQ*, which correspond to adipocytes (*Figure 6A*, left). Markers for endothelial cells (*CDH5*), and diverse markers for macrophage and monocytes (*CCR2*, *CD14*, *CD68*, and *ADGRE1*) could be detected in the corresponding cell types, but *MAOA* and *ADIPOQ* (as expected) transcripts were virtually undetectable in cells other than adipocytes. In mouse adipose tissues (*Figure 6A*, right), *Maoa* transcripts were detected at low levels in adipocytes, macrophages, and endothelial cells, with the highest level of transcript expression being detected in cells of the mesothelium. High levels of transcripts for *Adipoq* were detected in adipocytes, and various macrophage and monocyte markers could be detected in immune cell types. Both human and mouse adipocytes also contained transcripts for *SLC22A3*, the major extra-neuronal catecholamine transporter required for norepinephrine clearance (*Figure 6A*, bottom row). These results indicate species-specific, high-level expression of MAOA in human adipocytes. Parallel RT-PCR and western blotting analysis of human adipocytes during differentiation in vitro indicate that transcriptional levels of *MAOA* closely reflect protein expression (*Figure 6B*), suggesting that transcriptomic data reflect protein abundance.

To directly assess MAOA protein expression in adipocytes, we performed high resolution confocal imaging of human and mouse adipose tissues, and quantitative immunostaining of WIM and TIM. At lower magnification, sections of human adipose tissue displayed clear staining for MAOA, but no specific staining could be detected in numerous sections of inguinal mouse adipose tissue. In contrast, both tissues were clearly stained for the lipid droplet protein PLIN1 (*Figure 6C*). At higher resolution, MAOA could be seen localized between the lipid droplet, marked by perilipin staining, and the periphery of the cell, where it co-localized with mitochondrial HSP70, consistent with its reported

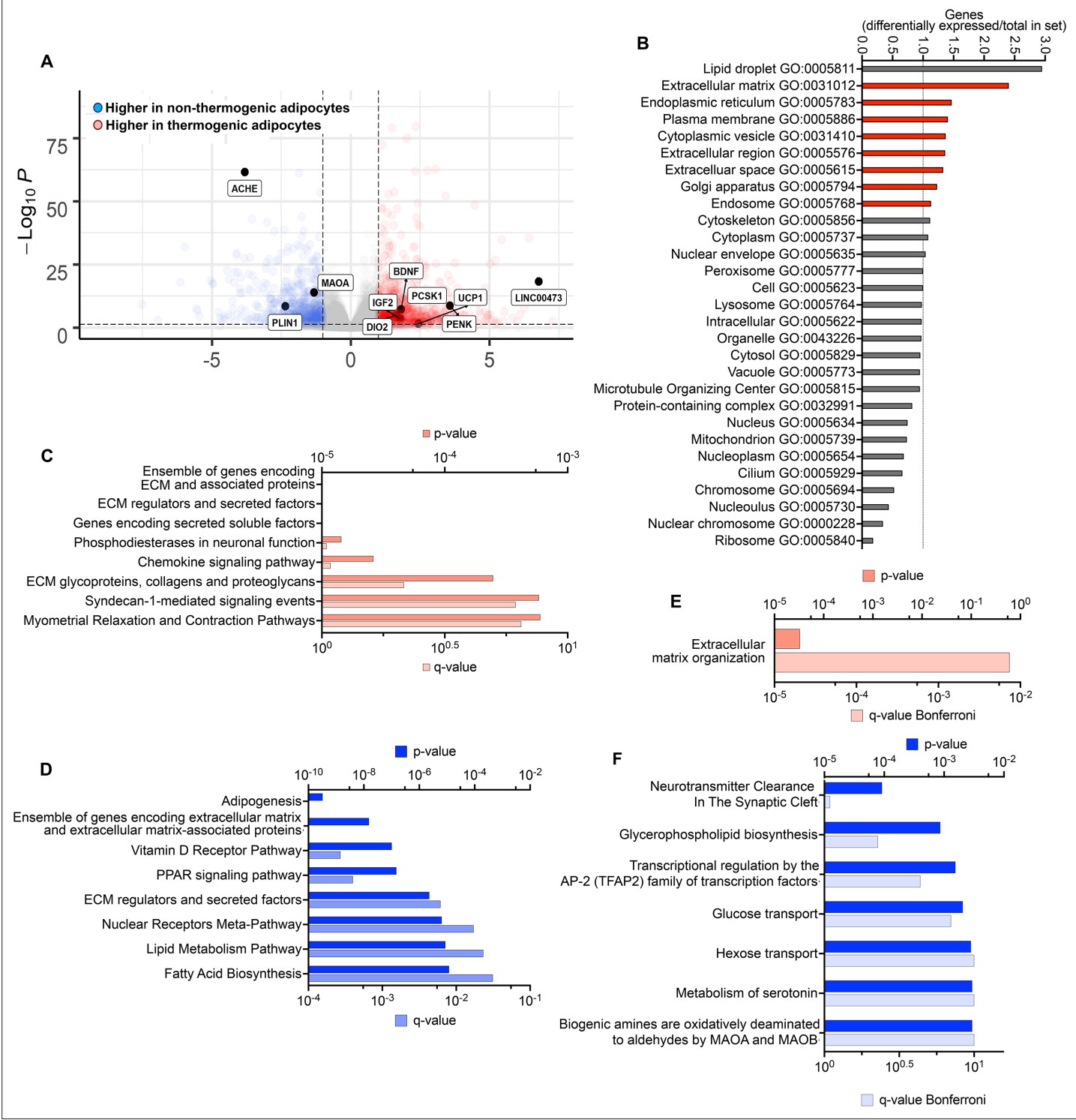

**Figure 5.** Differentially expressed genes persisting through tissue development. (**A**) Volcano plot of genes differentially expressed between non-thermogenic and thermogenic adipocytes prior to implantation. Highlighted are selected genes characteristic of human thermogenic adipocytes and genes involved in the control of neurotransmission. (**B**) Enrichment of GO categories by genes differentially expressed between non-thermogenic and thermogenic adipocytes, with categories for secretory proteins highlighted in red. (**C, D**) Pathway (REACTOME) enrichment analyses of genes more highly expressed in thermogenic (**C**) or non-thermogenic (**D**) adipocytes. E. Pathway analysis of genes more highly expressed in thermogenic adipocytes that remained more highly expressed in TIM. F. Pathway analysis of genes more highly expressed in non-thermogenic adipocytes that remained more highly expressed in WIM.

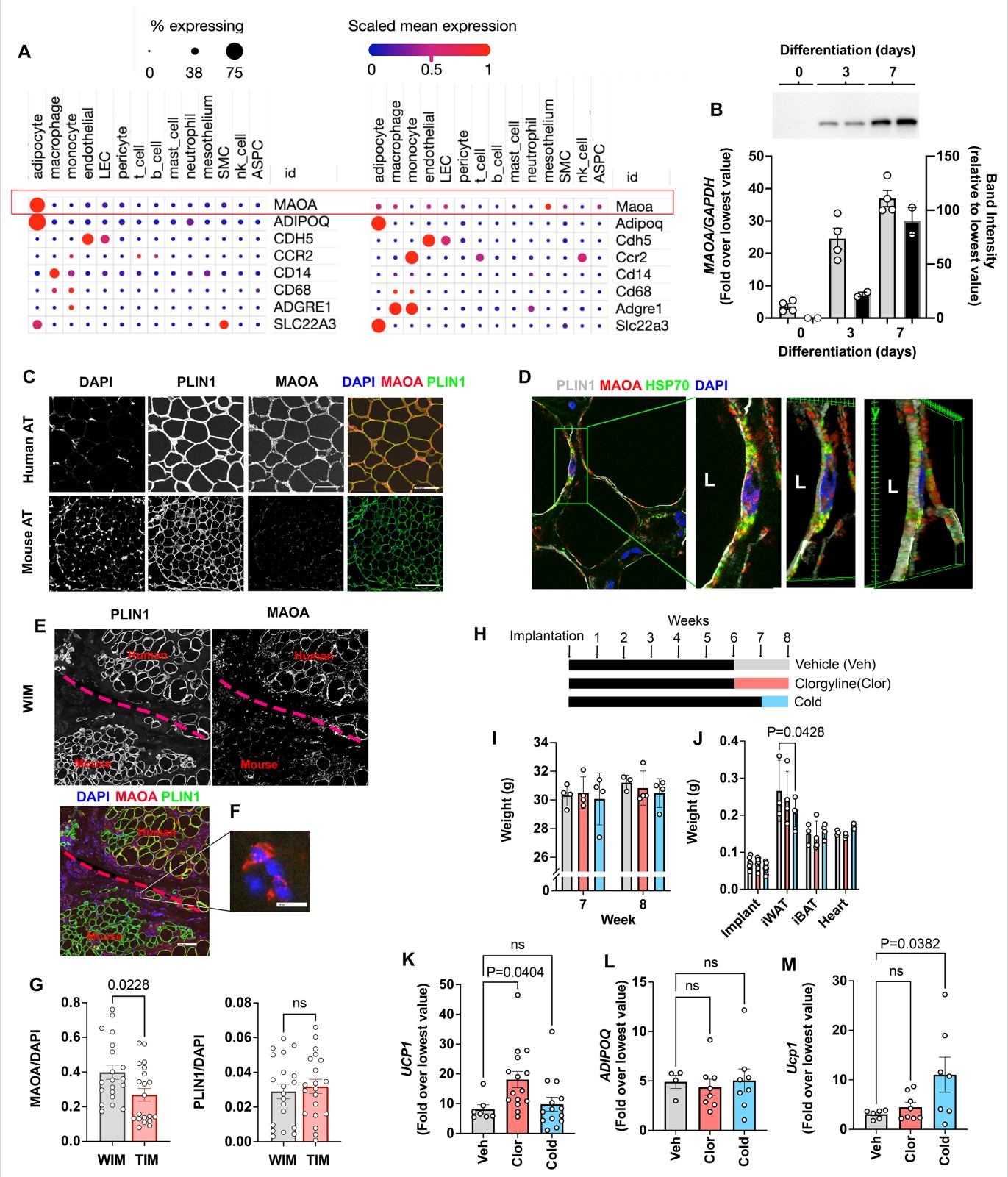

**Figure 6.** Species specific expression of MAOA in adipocytes. (**A**) Dot plots of indicated genes generated using data from *Emont et al., 2022*, available here (**B**) RT-PCR and western blotting at indicated days following induction of differentiation in human adipocytes. Samples were prepared from two independent wells and subjected to western blotting with an antibody to MAOA that recognizes the mouse and human proteins (CellSignaling #75330). The intensity of the single detected band was quantified, and values plotted in the right Y axis. RT-PCR results are plotted in the left Y axis.

*Figure 6 continued on next page*

*Figure 6 continued*

Symbols represent the fold difference over the lowest value in the dataset and bars the mean and SEM of n=4 independent cultures. (**C**) Representative images of human and mouse subcutaneous adipose tissue stained with antibodies to MAOA (CellSignaling #75330 for mouse, CellSignaling #73030 for human), and PLIN1, and counterstained with DAPI. Bars = 100 um. (**D**) Confocal images of human adipose tissue stained with antibodies to PLIN1, MAOA (CellSignaling #73030), and mtHSP70 and counterstained with DAPI. Projection of 15 images taken at 0.5 μm intervals (bar=10 um) and indicated segment in 3D projections illustrating localization of nucleus, mitochondria and MAOA to the region between the lipid droplet and the cell periphery (bars = 3 um). (**E**) WIM and vicinal mouse adipose tissue, stained for PLIN1, MAOA (CellSignaling #75330), and DAPI, with red dotted line indicating boundary between hybrid tissue and endogenous adipose tissue. Bar = 100um. (**F**) Immune cell showing staining for MAOA. Bar = 10 μm. (**G**) Intensity of MAOA (left panel) and PLIN1 (right panel) relative to DAPI in 5 independently analyzed regions from each of n=4 WIM and n=4 TIM thin sections, as described in Materials and methods. Statistical significance of the differences was assessed using two-tailed student t-tests, and exact values are shown. (**H**) Experimental timeline for exposure on mice harboring WIM to clorgyline or cold. (**I**) Weight of mice at 7- and 8 weeks (**J**) Weights of dissected tissues at 8 weeks. (**K**) RT-PCR for human *UCP1*. (**L**) RT-PCR for human *ADIPOQ*. For (K and L), bars represent the mean and S.E.M and symbols correspond to each independent implant (n=7–8 implants). M. RT-PCR for mouse *Ucp1* in excised inguinal adipose tissue (n=7–8 fat pads). Bars represent the mean and S.E.M and symbols correspond to each independent fat pad. (**I-M**) Statistical significance of the differences was assessed using one-way ANOVA with Dunett's multiple comparison correction, and exact values are shown.

The online version of this article includes the following source data and figure supplement(s) for figure 6:

**Source data 1.** Uncropped, labeled western blot.

**Figure supplement 1.** RT-PCR for additional thermogenic genes.

localization to the outer mitochondrial membrane (*Figure 6D*). Imaging of implants confirmed that MAOA protein is detected in human adipocytes within the implant, but not in vicinal endogenous mouse adipocytes (*Figure 6E*), despite being detectable in some sections in mouse immune cells infiltrating the implant (*Figure 6F*). To assess whether the lower expression of MAOA in TIM, detected by bulk RNA sequencing, was observable at the protein level, we stained thin sections of excised implants with antibodies to PLIN1 and MAOA, as well as DAPI to control for cellularity of the sections. MAOA staining was lower on average in sections from TIM compared to WIM (*Figure 6G*, left panel), while PLIN staining was not different between conditions (*Figure 6G*, right panel).

Monoamine oxidases are major mechanisms by which norepinephrine, epinephrin and serotonin are degraded (*Tipton, 2018*), and clorgyline, a specific, irreversible inhibitor of MAOA have been used in the treatment of depression for many years (*Lipper et al., 1979*). To determine whether expression of MAOA in human adipocytes might be functionally relevant, we treated mice harboring WIM with a low dose of the clorgyline for 2 weeks. A low dose was used to minimize its effects on endogenous brain and peripheral mouse Maoa. A separate cohort of mice was gradually acclimated to low ambient temperature as an alternative method to enhance adrenergic tone (*Figure 6H*). Neither treatment resulted in significant changes in body weights (*Figure 6I*), nor in weights of most tissues except for a decrease in inguinal subcutaneous adipose tissue weight in cold-acclimated mice (*Figure 6J*), which was expected given the reliance of BAT on peripheral lipolysis to sustain thermogenesis. However, implants from clorgyline-treated mice expressed significantly elevated *UCP1* (*Figure 6K*), with no change in *ADIPOQ* expression (*Figure 6L*). Effects on other genes associated with thermogenesis did not achieve statistical significance, potentially due to the low doses of clorgyline and mild cold exposure conditions employed (*Figure 6—figure supplement 1*). *Ucp1* in mouse subcutaneous adipose tissue was not affected by clorgyline, despite it being responsive to cold stimulation (*Figure 6M*), demonstrating that the effect of clorgyline was specific for human adipocytes.

## Discussion

Efforts to understand mechanisms of human thermogenic adipose tissue development are limited by the difficulty of accessing adipose tissue from human subjects over time, and from paucity of models to study human tissue development. Here, we leverage a species-hybrid model in which adipose tissue is formed after implantation of developing human adipocytes into mice. This model is enabled by methods to generate multipotent human progenitor cells that can be driven towards differentiation into multiple adipocyte subtypes (*Min et al., 2019*). Adipocytes complete maturation in vivo, developing key species-specific features, such as larger cell size compared to mouse adipocytes, and maintaining the thermogenic phenotype induced in-vitro. Developing adipocytes recruit host cellular

components, including vascular endothelial cells, adipose stem/progenitor cells, immune cells, and neuronal components to make functional adipose tissue.

An important question addressed by this work is whether supporting structures such as the vasculature of thermogenic adipose tissue differ qualitatively from that of non-thermogenic tissue. By distinguishing adipocytes from other cellular components at the transcriptional level and complementing these data with single cell atlases (*Gunda et al., 2014*; *Lamalice et al., 2007*), we find that vascular endothelial and other non-adipocyte cells of thermogenic adipose tissue, are qualitatively different. Vascularization of the developing tissue involves infiltration of mouse endothelial cells, followed by formation of mature blood vessels, as seen in other developing systems (*Lamalice et al., 2007*). Notably, after 8 weeks infiltrating mouse cells in thermogenic tissue form a denser vascular network, and display a significant increase in extracellular matrix elements (see *Supplementary file 1*) including *Eln*, *Fmod*, *Col5a3*, *Col8a2,* and *Col4a2*, the later which is associated with small vessel stability (*Gunda et al., 2014*). The composition of the extracellular matrix, and its resulting mechanical properties have been found to directly influence adipocyte development and thermogenic features (*Bauters et al., 2017*; *Lee et al., 2016*; *Takata et al., 2020*). Our results enable further experiments to explore the role of these specific extracellular matrix elements on human thermogenic adipocyte differentiation and function. Endothelial cells of the thermogenic tissue also express lower levels of circadian clock genes *Dbp*, *Per2,* and *Per3*, which are known to oscillate in adipose tissues of mice and humans (*Aggarwal et al., 2017*; *Dankel et al., 2014*; *Lekkas and Paschos, 2019*; *Otway et al., 2011*), and have been associated with the activity of thermogenic adipocytes (*Adlanmerini et al., 2019*; *Froy and Garaulet, 2018*). Our finding that these genes are expressed in the vasculature may explain the parallel oscillations of these genes in other peripheral tissues (*Soták et al., 2016*; *Yamamoto et al., 2004*).

In addition to a qualitative and quantitative different vasculature, thermogenic adipocytes also recruit a denser neuronal sympathetic network. The ability of implanted thermogenic adipocytes to recruit this neurovascular network is critical, as in the absence of external stimuli adipocytes lose their thermogenic phenotype within days. At 8 weeks of development thermogenic adipocytes express increased levels of genes associated with neurogenesis (*THBS4*, *TNC*, *NTRK3,* and *SPARCL1*) (*Gan and Südhof, 2020*; *Girard et al., 2014*; *Naito et al., 2017*; *Ramirez et al., 2021*; *Singh et al., 2016*; *Tucić et al., 2021*). Interestingly, two of these factors, Thbs4 and Sparcl1, have been identified as factors in blood from young mice that can directly stimulate synaptogenesis (*Gan and Südhof, 2020*). Beige adipocytes have been found to be neuroprotective (*Guo et al., 2021*), raising the possibility that these neurogenic factors produced by thermogenic adipocytes might not only enhance local innervation but might contribute to neuroprotection.

In addition to enhanced levels of neurogenic factors, we found that thermogenic adipocytes display decreased expression of neurotransmitter clearance enzymes MAOA and ACHE. MAOA catalyzes the oxidative deamination of norepinephrine and would be expected to affect thermogenesis by dampening adrenergic tone. ACHE is responsible for the catabolism of acetylcholine, which has been found to activate thermogenic adipocytes in the subcutaneous adipose tissue of mice (*Jun et al., 2020*). Importantly Maoa has been implicated in the control of adipose tissue thermogenesis in mice (*Camell et al., 2017*; *Pirzgalska et al., 2017*), through its expression in a specific subset of macrophages. Our studies indicate that in contrast to mice, human *MAOA* is strongly expressed in adipocytes, which also express the non-neuronal transporter *SLC22A3*, allowing for transport and degradation of catecholamines. These findings, which are consistent with those of others (*Bour et al., 2007*) suggest that adipocytes themselves play a major role in establishing adrenergic tone and controlling thermogenic activity in humans.

Correlations between human adipocyte MAOA and thermogenesis can be inferred from findings reporting that MAOA activity is increased in visceral adipose tissue of obese compared to non-obese patients (*Sturza et al., 2019*), which have also been reported to display decreased expression of UCP1 (*García-Alonso et al., 2016*; *Lim et al., 2020*; *Yu et al., 2018*). While more studies will clarify these associations, our studies directly demonstrate that inhibition of MAOA with low doses of clorgyline enhances expression of UCP1 in human but not mouse adipocytes, highlighting the functional relevance of the human adipocyte-expressed enzyme. It is notable that clorgiline induced UCP1 expression in the human tissue, while this was not seen in response to mild cold exposure. This could be attributable to the lower innervation density of WIM, and to a generally lower sympathetic

innervation in the flank of the mouse compared to inguinal and interscapular regions. Further studies will help determine whether the site of implantation affects responsiveness of hybrid tissue to cold exposure, and whether inhibition of MAOA has a synergistic effect. The finding that inhibition of MAOA can induce human adipocyte beiging in vivo suggests brain impenetrable, adipocyte favoring formulations of MAOA inhibitors as potential therapeutic agents in metabolic disease.

# Materials and methods

**Key resources table**

| Reagent type (species) or resource | Designation | Source or reference | Identifiers | Additional information |
|---|---|---|---|---|
| Cell line (*Homo sapiens*) | Adipose tissue progenitor cell | Corvera Laboratory | Mesenchymal progenitor cells | Cells used in this paper are generated from small fragments of surgically excised male or female human adipose tissue cultured in MatriGel, in the presence of normocin, penicillin and streptomycin. Cultures are recovered using dispase, expanded by two passages using trypsin, and frozen. Cells are then thawed and used for in experiments with no further passaging. Cells are not transformed, nor cultured beyond three passages and are therefore not routinely tested for mycoplasma nor subject to further authentication. Adipose tissue from which the primary cells are derived is obtained in accordance with the UMass Chan Institutional Review Board IRB ID 14734_13 |
| Antibody | Anti-Perilipin-1 (Rabbit monoclonal) | Cell Signalling | #9349 | IF(1:500), |
| Antibody | Anti-MAOA (Rabbit monoclonal) | Cell Signalling | #73030 | IF(1:500) WB (1:500) |
| Antibody | Anti-MAOA (Rabbit monoclonal) | Cell Signalling | #75330 | IF(1:500) WB (1:500) |
| Antibody | Anti-Tyrosine hydroxylase (Rabbit polyclonal) | Millipore Sigma | #AB152 | IF (1:500) |
| Antibody | Anti-CD45 (Mouse monoclonal) | Abcam | #282747 | IF (1:500) |
| Antibody | Anti-F4/80 (Rat monoclonal) | Abcam | #6640 | IF (1:500) |
| Commercial assay, kit | Adiponectin ELISA | Invitrogen | #KHP0041 | Serum diluted 1:50 |
| Other | Isolectin GS-IB4 AlexaFluor-647 conjugate | Invitrogen | #I32450 | Staining, 1:200 |
| Other | Ulex Europaeus Agglutinin I DyLight 594 | Vector Laboratories | #DL-1067–1 | Staining, 1:200 |
| Other | DAPI stain | Invitrogen | #D1306 | Counterstaining 1 µg/mL |

## Human Subjects

Abdominal adipose tissue was collected from de-identified, discarded tissue of patients undergoing panniculectomy, with approval from the University of Massachusetts Institutional Review Board (IRB ID 14734_13).

## Generation of cells and implants

Small pieces of fat (~1 mm³) from the excised human adipose tissue were embedded in Matrigel (200 explants/10 cm dish) and cultured for 14 days as described before (*Rojas-Rodriguez et al., 2019*). Single- cell suspensions were obtained using dispase and plated into 150 mm tissue culture dishes.

After 72 hr, cells were split 1:2, grown for an additional 72 hr, recovered by trypsinization and frozen. To generate the implants, ~10E7 cells were thawed into each 150 mm plate, and after 72 hr split 1:2 at a dense seeding density of ~8E6 cells per 150 mm plate. Upon confluence (approximately 72 hr after plating), differentiation was induced by replacing the growth media with DMEM +10% FBS, 0.5 mM 3-isobutyl-1- methylxanthine, 1 µM dexamethasone, and 1 µg/mL insulin (MDI). MDI media was changed daily for 72 hr at which time the differentiation medium was replaced by DMEM +10% FBS, 50% of which was replaced every 48 hr for 10 days. After 10 days of differentiation, a subset of cells was stimulated daily for 7 days with forskolin (1 µM final concentration) to induce the thermogenic phenotype. Single-cell suspensions of differentiated and thermogenic-induced adipocytes were obtained by incubation for 7–10 min in Trypsin (1 X)/collagenase (0.5 mg/mL). Proteases were quenched by dilution into culture media, cells pelleted by centrifugation at 500 rpm for 10 min.

The media layer between floating and pelleted cells was removed, and remaining cells brought to 1 mL total volume with ice-cold PBS, placed on ice and mixed with an equal volume of ice-cold Matrigel. 0.5 mL aliquots of the cell suspension were placed in 1 ml syringes on ice and injected subcutaneously into each flank of immunodeficient male nude mice, 8–10 weeks old, (Nu/J Jackson labs stock no: 002019) using a 20 G needle.

A cohort of mice were randomized at 6 weeks post-implantation to either pharmacological inhibition of MAOA, or cold exposure. For MAOA inhibition, mice received intraperitoneal injections of the MAOA-specific irreversible inhibitor Clorgyline (Sigma-Aldrich Cat M3778) at a dose of 1 mg/kg diluted in PBS. Control mice were treated with the same concentration of DMSO in PBS. Injections were administered 5 days on – 2 days off at the same time every day for a total of 2 weeks. For cold exposure, animals were transferred to cold chambers and gradually acclimated from 22°C to 16°C over a period of 3 days. Mice remained at 16 °C for 2 additional days. At the end of the 8-week time-point, all the animals were sacrificed, and tissues were collected for analysis.

At each experimental time point, mice were sacrificed, and developed implants were dissected and snap frozen until further analysis. All procedures were performed in accordance with the University of Massachusetts Medical School's Institutional Animal Care and use Committee protocol PROTO202100015.

## Human-specific adiponectin ELISA

Blood was obtained through cardiac puncture, centrifuged, and the plasma was collected and stored at –80 °C. A human-specific adiponectin ELISA (Invitrogen KHP0041) was used to measure the concentration of human adiponectin in mouse blood.

## RNA extraction and quantitative PCR

Implants were placed on TRIzol (Invitrogen) and homogenized with Tissuelyser (Qiagen). Total RNA was reverse transcribed using the iScript cDNA Synthesis Kit (Bio-Rad) per manufacturer's protocol. Either SYBR Green or PrimePCR Probes (Bio-Rad) were used to detect amplification of human- and mouse-specific genes using the following primers:

| Primers | Forward | Reverse |
| --- | --- | --- |
| Human RPL4 | GCCTGCTGTATTCAAGGCTC | GGTTGGTGCAAACATTCGGC |
| Mouse Rpl4 | CCCCTCATATCGGTGTACTCC | ACGGCATAGGGCTGTCTGT |
| Human PLIN1 | ACCAGCAAGCCCAGAAGTC | CATGGTCTGCACGGTGTATC |
| Mouse Plin1 | CTGTGTGCAATGCCTATGAGA | CTGGAGGGTATTGAAGAGCCG |
| Human CDH5 | CTGCTGCAGGGTCTTTTTCT | AGGGCATGATGGTCAGTCTC |
| Mouse Cdh5 | TACTCAGCCCTGCTCTGGTT | TGGCTCTGTGGTGCAGTTAC |
| Human UCP1 | AAGTCCAAGGTGATTGCC | TTACCACAGCGGTGATTGTTC |
| Mouse Ucp1 | GTGAACCCGACAACTTCCGAA | TGCCGAGCAAGCTGAAACTC |

## RNA-Seq

RNA was extracted from white and thermogenic implants using the TRIzol method. Library preparation was performed using the TruSeq cDNA library construction (Illumina). Samples were processed on the Illumina HiSeq 550 sequencing system with the NextSeq 500/550 High Output kit v2.5 (Illumina, Cat. No. 20024906). The generated fastq files were loaded into the DolphinNext platform (https://dolphinnext.umassmed.edu/) and the Bulk RNA sequencing pipeline was used. The. fastq files were aligned to both the human (hg38) and the mouse (mm10) genome. The resulting alignments were processed using the R-package XenofilteR to classify reads as either of human or mouse origin. Reference (https://github.com/PeeperLab/XenofilteR; *Netherlands Cancer Institute - Genomics Core Facilty, 2022*; *Kluin et al., 2018*) for more details on XenofilteR source code. Once aligned, the files were run through RSEM for normalization. Differential expression analysis was performed using the DEBrowser platform (https://debrowser.umassmed.edu/). Gene ontology analysis was performed by combining results from TopFunn (https://toppgene.cchmc.org/).

## Single-cell RNA-Seq processing

Single-cell RNA sequencing data was obtained from data from *Sun et al., 2020* (https://www.ebi.ac.uk/arrayexpress/experiments/E-MTAB-8564/) and (*Burl et al., 2018*) (https://www.ncbi.nlm.nih.gov/sra/?term=SRP145475). The provided.fastq files were loaded into DolphinNext and processed using the Cell Ranger pipeline. The output files from Cell Ranger were then loaded into the Seurat (https://satijalab.org/seurat/index.html). We removed unwanted cells from the data by sub-setting features as follow: nFeature_RNA >100 & nFeature_RNA <5,000. We then proceeded to normalize the data using a logarithmic normalization method with a scale factor of 10,000. We then applied a linear transformation to scale the data followed by dimension reduction analysis (PCA). Following PCA analysis we determined that the first 8 principal component analysis were optimal for further analysis. To cluster the cells, we performed non-linear dimension reduction analysis using the UMAP algorithm. Analysis was done using the first 8 principal components and a resolution of 0.3. Finally, differentially expressed genes for each cluster were identified to determine biomarkers for cell cluster identification.

## Cell predictor using DWLS deconvolution

Source code for DWLS deconvolution can be found at https://github.com/dtsoucas/DWLS, (*Solivan-Rivera, 2022* copy archived at swh:1:rev:b73dd13d915575f5becf3cdd1a566877e73605d7). Briefly, metadata from the Seurat object was used to generate a matrix containing the reads from the scRNA and one containing the cluster labels for each gene. A signature matrix was generated using the MAST function provided with the DWLS deconvolution code. Only genes in common between the scRNA and the bulk RNA dataset were used for prediction. DWLS deconvolution was performed using the solveDampenedWLS function and proportions of predicted cells were plotted.

## Single nuclei RNA-Seq processing

Seurat objects for single-nuclei RNA sequencing was kindly provided by the Wolfrum lab, who published their analysis (*Sun et al., 2020*). Briefly, the Seurat objects containing the raw data were loaded the Seurat package, and two different samples were processed: single nuclei from human brown adipose tissue, single nuclei from mouse brown adipocytes. Briefly, for the human brown adipose tissue dataset, we first removed unwanted cells from the data by sub-setting features as follow: nCount_RNA >1000 & nCount_RNA <8000 & prop. mito <0.3. We then normalized the data using a logarithmic normalization with a scale factor of 10000. We followed this by applying a linear transformation to scale the data followed by dimension reduction analysis (PCA). We determined that the first 10 principal component analysis were optimal for further analysis. To cluster the cells, we performed non-linear dimension reduction analysis using the UMAP algorithm. Analysis was done using the first 8 principal components and a resolution of 0.3. For the mouse brown adipocytes, we first removed unwanted cells from the data by sub-setting features as follow: nFeature_RNA >1000 & nFeature_RNA <5000. We then normalized the data using a logarithmic normalization with a scale factor of 10,000. We followed this by applying a linear transformation to scale the data followed by dimension reduction analysis (PCA). We determined that the first 8 principal component analysis were optimal for further analysis. To cluster the cells, we performed non-linear dimension reduction analysis using the UMAP algorithm. Analysis was done using the first 8 principal components and a

resolution of 0.2. Finally, clusters names were assigned using SingleR (https://github.com/dviraran/SingleR; *dviraran, 2020*; *Aran et al., 2019*). The human primary cell atlas (*Mabbott et al., 2013*) was used as a reference to identify clusters based on differentially expressed genes of human brown adipose tissue.

## Histochemistry and quantification

All samples were fixed in 4% paraformaldehyde overnight at 4 °C and thoroughly washed with PBS. Tissue sections (8 μm) were mounted on Superfrost Plus microscope slides (Fisher Scientific) and stained with hematoxylin and eosin. For whole-mount staining, tissue fragments (~1 mm³) were stained and mounted between 1.5 mm coverslips sealed with ProLong Gold Antifade Reagent (Life Technologies). Mouse vasculature was detected using isolectin GS-IB4 AlexaFluor-647 conjugate (Invitrogen I32450) and human vasculature using Ulex Europaeus Agglutinin I (UEA I), DyLight 594 (Vector Laboratories DL-1067–1). Nuclei were stained using DAPI (Life Sciences 62249). Perilipin-1 was detected with an antibody that recognizes both human and mouse protein (CellSignaling #9349). MAOA was detected using CellSignaling #73030 (human) or CellSignaling #75330 (human and mouse). To identify sympathetic nerves an antibody for Tyrosine Hydroxylase was used (Millipore Sigma - AB152). Macrophages were identified by co-staining using antibodies for CD45 (Abcam – ab282747) and F4/80 (Abcam – ab6640). Mitochondrial HSP70 was detected using Invitrogen MA3-028. Tissue sections were imaged using ZEISS Axio Scan Z1. Whole-mount images were acquired using an Olympus IX81 microscope (Center Valley, PA) with dual Andor Zyla sCMOS 4.2 cameras (Belfast, UK) mounted on an Andor TuCam two camera adapter (Belfast, UK). To prevent visual biases, entire sections were imaged using a scanning microscope.

Image analysis was performed using the open software platform FIJI. A single montage containing all images was generated for all channels analyzed, thereby subjecting all images to the same filtering and quantification processes. To measure adipocyte sizes, images were converted to 8-bit grayscale, binarized using the default Threshold function, eroded twice, and object number and size measured through particle analysis (size = 350–2900 μm, circularity = 0.25–1). Density distributions (scaled) were generated using R-studio through the density distribution function in ggplot2. To measure vascular density and size, isolectin stained images were background subtracted (rolling radius = 1), binarized, and subjected to particle analysis. The total vascular area of each section was the sum of the individual particles. The average vascular size was the mean of the individual particles. For macrophage quantification, CD45, F4/80 and DAPI images were background subtracted (rolling radius = 1) and binarized. To isolate macrophages that were both CD45 and F4/80 positive, we used the image calculator function (image calculator >operation: AND). Particle analysis was performed, and the number of particles were normalized to the number of nuclei present in the whole section to account for differences in section sizes. Sympathetic density was calculated using TH and isolectin stained images after background subtraction (rolling radius = 1) and binarization. Total sympathetic area in the section was the sum of all particles normalized to the number of nuclei present in the whole section to account for differences in section sizes. Images were analyzed by two investigators independently of each other, blinded to the nature of the samples.

To quantify MAOA levels between WIM and TIM, thin sections from n=4 each explant were stained for PLIN1, MAOA and DAPI. Whole sections were imaged using a TissueFAX SL tissue cytometer using ×20 magnification. Images were collated using the TissueFAXS 7 viewer and imported into FIJO for further analysis. Thresholds were applied to all sections in each channel equally, and MAOA and PLIN1 intensities were divided by the DAPI intensity in each region to account for cellularity. To account for differences in section sizes, 5 equally sized subsections of each image were analyzed and used to determine differences between WIM and TIM.

## Thermogenic withdrawal of cultured adipocytes

Cells (~6.0 × 10⁶) were thawed and seeded on 6-well plates, differentiated into white adipocytes, and stimulated with forskolin as described above. After chronic forskolin stimulation, a subset of cells was removed from daily stimulation for 10 days. Images were taken and samples were placed on TRIzol (Invitrogen) at days 0, 2, 5, 7, and 10 of withdrawal. Samples were homogenized using the Tissuelyser (QIAGEN), and total RNA was then isolated for analysis as described above.

## Materials availability

The count data and processed data for bulk RNASeq has been deposited in the Gene Expression Omnibus under the accession number GSE200141.

## Statistical analysis

Sample sizes ranged from five to seven biological replicates for each experiment and were based on variance associated with each parameter tested as assessed in prior studies (*Min et al., 2016*; *Rojas-Rodriguez et al., 2019*; *Rojas-Rodriguez et al., 2020*) Statistical analysis was performed using GraphPad Prism 9. Statistical tests and exact p values are described in each figure legend. Data were tested for normality before use of parametric tests, and when normality could not be verified, non-parametric tests were used. Statistical significance between groups was estimated using ordinary one-way ANOVA corrected for multiple comparisons as described in each figure legend.

## Acknowledgements

This study was supported by NIH grants DK089101 and DK123028 to SC and GM135751 to JSR. We acknowledge the use of services from the UMASS SCOPE core for high resolution confocal imaging, the biomedical imaging (BIG) facility for epifluorescence imaging and image analysis consultation. Finally, we acknowledge the use of services from the UMASS Morphology Core facility for sample preparation and sectioning for histological analysis.

## Additional information

### Funding

| Funder | Grant reference number | Author |
| --- | --- | --- |
| National Institutes of Health | DK089101 | Silvia Corvera |
| National Institutes of Health | DK123028 | Silvia Corvera |
| National Institutes of Health | GM135751 | Javier Solivan-Rivera |

The funders had no role in study design, data collection and interpretation, or the decision to submit the work for publication.

### Author contributions

Javier Solivan-Rivera, Conceptualization, Software, Investigation, Visualization, Methodology, Writing – original draft, Writing – review and editing; Zinger Yang Loureiro, Raziel Rojas-Rodriguez, Formal analysis, Methodology, Writing – review and editing; Tiffany DeSouza, Conceptualization, Formal analysis, Investigation, Methodology, Writing – review and editing; Anand Desai, Conceptualization, Investigation, Visualization, Methodology, Writing – review and editing; Sabine Pallat, Conceptualization, Investigation, Visualization; Qin Yang, Investigation, Methodology; Rachel Ziegler, Methodology, Writing – review and editing; Pantos Skritakis, Conceptualization, Visualization, Methodology, Writing – review and editing; Shannon Joyce, Conceptualization, Methodology, Writing – review and editing; Denise Zhong, Conceptualization, Writing – review and editing; Tammy Nguyen, Conceptualization, Writing – original draft, Writing – review and editing; Silvia Corvera, Conceptualization, Software, Formal analysis, Funding acquisition, Investigation, Visualization, Methodology, Writing – original draft, Project administration, Writing – review and editing

### Author ORCIDs

Javier Solivan-Rivera http://orcid.org/0000-0002-3238-485X
Zinger Yang Loureiro http://orcid.org/0000-0001-8543-4841
Silvia Corvera http://orcid.org/0000-0002-0009-4129

## Ethics

All procedures were performed in accordance with the University of Massachusetts Medical School's Institutional Animal Care and use Committee protocol PROTO202100015.

## Decision letter and Author response

Decision letter https://doi.org/10.7554/eLife.78945.sa1
Author response https://doi.org/10.7554/eLife.78945.sa2

# Additional files

## Supplementary files

• Supplementary file 1. Top 50 Differentially expressed mouse genes between WIM and TIM.

• Supplementary file 2. Differential expression of human genes between WIM and TIM.

• Supplementary file 3. Top 50 genes differentially expressed between non-thermogenic and thermogenic adipocytes prior to implantation.

• Supplementary file 4. Genes differentially expressed between non-thermogenic and thermogenic adipocytes that are maintained differentially expressed in WIM and TIM.

• Transparent reporting form

## Data availability

RNASeq data has been deposited in the Gene Expression Omnibus under the accession number GSE200141.

The following dataset was generated:

| Author(s) | Year | Dataset title | Dataset URL | Database and Identifier |
|-----------|------|---------------|-------------|-------------------------|
| Corvera S | 2022 | A neurogenic signature involving monoamine oxidase-a controls human thermogenic adipose tissue development | https://www.ncbi.nlm.nih.gov/geo/query/acc.cgi?acc=GSE200141 | NCBI Gene Expression Omnibus, GSE200141 |

The following previously published datasets were used:

| Author(s) | Year | Dataset title | Dataset URL | Database and Identifier |
|-----------|------|---------------|-------------|-------------------------|
| Sun W | 2020 | Single nucleus RNA sequencing of human deep neck BAT | https://www.ebi.ac.uk/arrayexpress/experiments/E-MTAB-8564/ | ArrayExpress, E-MTAB-8564 |
| Burl RB | 2020 | Single cell RNA-sequencing of male C57BL/6j mouse epididymal white adipose tissue lineage marker negative stromal/vascular cells, 3 days CL treatment | https://www.ncbi.nlm.nih.gov/sra/SRX4074089 | NCBI Sequence Read Archive, SRX4074089 |

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
