## [Editor Report]

This paper provides important new information that will be of wide interest in the fields of metabolism and diabetes research. The authors implanted human adipocyte progenitors is potentially a novel approach to analyze the development of human thermogenic adipose tissue.

---

## [Decision Letter]

**Decision letter after peer review:**

Thank you for submitting your article "A Neurogenic Signature Involving Monoamine Oxidase-A controls Human Thermogenic Adipose Tissue Development" for consideration by *eLife*. Your article has been reviewed by 2 peer reviewers, and the evaluation has been overseen by a Reviewing Editor and Ma-Li Wong as the Senior Editor. The reviewers have opted to remain anonymous.

Essential revisions:

1. The description of Figure 1E-L (e.g., WIM and TIM) was missing in the main text. This is also the case in Figure 2E-J: the description was not seen in the main text. Thus, this reviewer could not appreciate the full context of the work.

2. The staining of MAOA is not clear (Figure 6E). MAOA is located at the out membrane of mitochondria while the data looks that MAOA is located at the plasma membrane. MAOA western blots should also be shown in WIM and TIM.

3. The paper only provides data on UCP1 mRNA to demonstrate that inhibition of MAOA induces browning of human white adipose tissue. Other thermogenic genes Cidea, COX8b, Ppargc1b, FABP3, et al., should be shown along with UCP1 and Cidea protein levels by Western blot.

4. In Figure 4, the authors define cell types by comparing the expression of target genes based upon previously published single-cell data. However, single-cell data is quite shallow and only reflects a fraction of gene expression biased to highly expressed genes whereas deep sequencing can determine even low expression levels. This makes these analyses somewhat problematic as the authors are using different biological samples.

5. In the paragraph “The neurovascular network differentially responds to implanted adipocytes and supports TIM thermogenic phenotype”, the data shown do not support this statement. No data of the neurovascular network of mouse adipose tissue transplanted with WIM or TIM are shown, instead showing whiter (less UCP1 level) subcutaneous adipose tissue in mouse implanted with TIM than WIM. Is there less neurovascular network in mouse implanted with TIM than WIM? How to explain?

6. One of the notable findings was that human adipocytes expressed MAOA. Although the authors used an MAOA inhibitor, a stringent approach to test the cell-autonomous role of MAOA in adipocytes would be to implant MAOA-depleted cells into mice.

7. An intriguing observation is the upregulation of ECM genes in thermogenic implanted fat. The authors want to discuss if ECM accumulation is involved in the activation or maintenance of adipose tissue thermogenesis.

8. In Figure 3H, the authors speculated the feedback mechanism of implanted fat to host thermogenesis. The authors should explore if there is an inverse correlation between thermogenesis in implanted fat vs. endogenous fat in the host.

9. The rationale for studying LINC00473 (in Figure 3E) is lacking. It is unclear why the authors studied this gene. Any functional role for adipose tissue development?

10. In Figure 6J, it is a little puzzling that the authors failed to find an increase in UCP1 expression in transplanted adipose tissue after cold exposure. The authors wish to provide explanations for the observation.

*Reviewer #2 (Recommendations for the authors):*

This manuscript uses a species-hybrid model in which functional human white and thermogenic adipose tissue develop in nude mice. Interestingly, human adipose tissue is fully able to recruit mouse vascular networks and sympathetic innervation during human adipocyte development. Compared to human white adipose tissue, human thermogenic adipose tissue recruited a denser mouse vascular network and denser sympathetic neuron innervation. The thermogenic adipose tissue was found to express higher neurogenesis genes but fewer neurotransmitter clearance genes, for example, MAOA (Monoamine oxidase A gene). Moreover, inhibition of MAOA with inhibitor-induced browning of human white adipose tissue. Although these findings provide novel and valuable information about the development of human thermogenic adipose tissue, there are a variety of issues that need to be addressed.

---

## [Author Response]

Essential revisions:1. The description of Figure 1E-L (e.g., WIM and TIM) was missing in the main text. This is also the case in Figure 2E-J: the description was not seen in the main text. Thus, this reviewer could not appreciate the full context of the work.

We apologize for this, it seems the final paragraphs of some pages were deleted when inserting line numbers. The revised version has been checked repeatedly to insure the content is complete.

2. The staining of MAOA is not clear (Figure 6E). MAOA is located at the out membrane of mitochondria while the data looks that MAOA is located at the plasma membrane. MAOA western blots should also be shown in WIM and TIM.

Adipocytes have a unique morphology where all cytoplasmic elements are restricted to a very thin rim between the lipid droplet and the plasma membrane. We have now included high resolution confocal images and counter-staining with mitochondrial markers to illustrate the co-localization of MAOA with mitochondria in a new panel (Figure 6D). We do not have sufficient material to enable quantitative western blotting from excised implants. However, we have performed a quantitative analysis of MAOA immunostaining from multiple sections of n=4 implants for each condition. The results are now included in Figure 6G, and support the findings derived from bulk RNA sequencing reported previously

3. The paper only provides data on UCP1 mRNA to demonstrate that inhibition of MAOA induces browning of human white adipose tissue. Other thermogenic genes Cidea, COX8b, Ppargc1b, FABP3, et al., should be shown along with UCP1 and Cidea protein levels by Western blot.

We have now included additional genes (*CIDEA, DIO2, LINC00473*) and present the results in Figure 6_FigureSupplet_1. We do not have sufficient material to perform quantitative western blots from the implants, unfortunately.

4. In Figure 4, the authors define cell types by comparing the expression of target genes based upon previously published single-cell data. However, single-cell data is quite shallow and only reflects a fraction of gene expression biased to highly expressed genes whereas deep sequencing can determine even low expression levels. This makes these analyses somewhat problematic as the authors are using different biological samples.

We agree with this criticism, and have now presented the results from a more recent analysis which harmonizes single-cell and single nuclear sequencing from mouse and human adipose tissues to compare expression of MAOA with that of other cell type markers. We also clarify the limitations inherent to this analysis.

5. In the paragraph "The neurovascular network differentially responds to implanted adipocytes and supports TIM thermogenic phenotype", the data shown do not support this statement. No data of the neurovascular network of mouse adipose tissue transplanted with WIM or TIM are shown, instead showing whiter (less UCP1 level) subcutaneous adipose tissue in mouse implanted with TIM than WIM. Is there less neurovascular network in mouse implanted with TIM than WIM? How to explain?

The reviewer is correct that the comparison is between the development of the neurovascular network innervating the tissue formed from implanted cells. We now clarify this in the text, and have changed the title of the sections to conform better to the results shown (“Development and maintenance of TIM thermogenic phenotype is not cell autonomous”). We refer to other findings reporting conditions in which the total content of thermogenic tissue in the organism appears to be sensed and balanced (“Schulz, T.J., et al., Brown-fat paucity due to impaired BMP signalling induces compensatory browning of white fat. Nature, 2013. 495(7441): p. 379-83.). However, elucidating these mechanisms is beyond the scope of this manuscript.

6. One of the notable findings was that human adipocytes expressed MAOA. Although the authors used an MAOA inhibitor, a stringent approach to test the cell-autonomous role of MAOA in adipocytes would be to implant MAOA-depleted cells into mice.

We agree, and are developing technology to enable deletion of the gene in implanted cells. However, we feel that the inhibitor used is highly specific and has been used clinically for many years, and the finding of a specific effect of the inhibitor on adipose tissue formed from human cells, but not in the endogenous mouse adipose tissue, argues strongly for a functional role for human adipocyte MAOA.

7. An intriguing observation is the upregulation of ECM genes in thermogenic implanted fat. The authors want to discuss if ECM accumulation is involved in the activation or maintenance of adipose tissue thermogenesis.

This is an excellent point and we have expanded the discussion on this finding, to include references to the role of the extracellular matrix in thermogenic adipocyte development and function in paragraph #2 of the Discussion section.

8. In Figure 3H, the authors speculated the feedback mechanism of implanted fat to host thermogenesis. The authors should explore if there is an inverse correlation between thermogenesis in implanted fat vs. endogenous fat in the host.

The finding that the amount of endogenous thermogenic fat seems to be balanced by endogenous feedback mechanisms has been described by others. We reference this finding (“Schulz, T.J., et al., Brown-fat paucity due to impaired BMP signalling induces compensatory browning of white fat. Nature, 2013. 495(7441): p. 379-83.), but feel that elucidating the underlying mechanisms is beyond the scope of the current manuscript.

9. The rationale for studying LINC00473 (in Figure 3E) is lacking. It is unclear why the authors studied this gene. Any functional role for adipose tissue development?

LINC00473 was identified in prior work (Tran, K.V., et al., Human thermogenic adipocyte regulation by the long noncoding RNA LINC00473. Nat Metab, 2020. 2(5): p. 397-412) as being highly correlated with human thermogenic fat abundance, and we have used it as a marker of thermogenic adipocytes. The reference to this was missing from the manuscript due to formatting issues, but has now been corrected.

10. In Figure 6J, it is a little puzzling that the authors failed to find an increase in UCP1 expression in transplanted adipose tissue after cold exposure. The authors wish to provide explanations for the observation.

There are several potential reasons for the lack of effect. The major one we feel is the localization of the developed tissue to the flank of the mouse. It is possible that this region is less well innervated compared to the inguinal subcutaneous region or the interscapular region where mouse thermogenic fat is localized. Colds exposure would activate through sympathetic innervation, which may not be sufficient in WIM. We have now addressed this point in the last paragraph of the discussion.